# Current Practices for U.S. Newborn Screening of Pompe Disease and MPSI

**DOI:** 10.3390/ijns6030072

**Published:** 2020-09-02

**Authors:** Elizabeth G. Ames, Rachel Fisher, Mary Kleyn, Ayesha Ahmad

**Affiliations:** 1Division of Pediatric Genetics, Metabolism and Genomic Medicine, Department of Pediatrics, University of Michigan Health System, D5240 Medical Professional Building, 1500 E. Medical Center Dr, Ann Arbor, MI 48109, USA; firachel@med.umich.edu (R.F.); ayeshaah@med.umich.edu (A.A.); 2Newborn Screening Follow-up Section, Michigan Department of Health and Human Services, 333 South Grand Avenue, Lansing, MI 48933, USA; kleynm@michigan.gov

**Keywords:** Lysosomal storage disease, NBS, early diagnosis, MPSI, Hurler syndrome, Hurler-Scheie syndrome, LOPD, IOPD

## Abstract

Two lysosomal storage disorders (LSDs), Pompe disease and Mucopolysaccharidosis type I (MPSI) were added to the Recommended Uniform Screening Panel (RUSP) for newborn screening (NBS) in 2015 and 2016, respectively. These conditions are being screened with variable practice in terms of primary and reflex analytes (either biochemical or molecular testing) as well as collection of short- and long-term follow-up elements. The goal of this study is to evaluate practices of state health departments in regards to screening methods and follow-up data collected. We conducted online surveys and phone questionnaires to determine each U.S. state’s practices for screening and follow-up of positive newborn screens. We report the first snapshot of practices for NBS for the LSDs included on the RUSP. All 50 U.S. states responded to our survey. The majority of U.S. states are not currently screening for Pompe disease and MPSI as of March 2020, but this number will increase to 38 states in the coming 1–3 years based on survey results. Our survey identifies data elements used by state health departments for short-and long-term follow-up that could serve as the basis of common elements for larger, public health-based analyses of the benefits and efficacy of screening for Pompe disease and MPSI.

## 1. Introduction

Newborn screening (NBS) for two lysosomal storage disorders (LSDs), Pompe disease and MPSI, were recently added to the recommended uniform screening panel (RUSP) in the United States [1,2] based on effective treatment options [3,4,5,6,7]. Most NBS laboratories utilize enzyme activity measurements from dried blood spots, but due to high rates of pseudodeficiency alleles and false positives [8,9], reflex testing of initial positive results has been proposed for many LSDs [10,11]. Pseudodeficiency is a phenomenon seen in many LSDs where in vitro measurement of enzymatic activity (utilizing artificial substrates) appears greatly reduced, but the individual remains clinically unaffected due to normal in vivo enzymatic activity. Pseudodeficiency alleles are molecular variants that lead to this abnormal in vitro enzymatic testing result [12,13]. It is important to acknowledge the differences in timing between NBS-related follow-up for Pompe and MPSI. Infantile-onset Pompe disease requires emergent treatment within the first two weeks of life for best outcomes [5,14]. While early treatment for severe MPSI has also been shown to stop the accumulation of CNS damage, the timeline for hematopoietic stem cell transplantation is on the order of weeks or months rather than days [15]. Detection of late-onset forms of Pompe disease by NBS have been well-documented [6,16], but the degree that attenuated forms of MPSI are detected by NBS are less clear [17,18].

Each U.S. state has different screening protocols for NBS including choice of primary analyte, use of reflex testing, type of reflex testing, and degree of follow-up for positive NBS. As we embark on screening for these disorders and additional LSDs are considered for NBS, it is important to capture the early efforts of NBS. We conducted a survey of individuals from state NBS laboratories, state health department follow-up programs, or clinical geneticists involved in NBS follow-up. We report responses from all 50 U.S. states about their current and planned practices.

## 2. Materials and Methods

An online survey (Qualtrics, Provo, UT, USA) was sent to all NBS state laboratories. The survey was voluntary and was deemed exempt and not regulated by the University of Michigan’s IRB committee (HUM00152502). The survey was distributed through email list serves to NBS labs, as QR codes at MPSI Newborn Screening Educational Meeting at the University of Minnesota (25–27 September 2019), and through personal contacts in individual states. Some states were contacted by phone and surveys were collected verbally. Anonymity for this publication was maintained by presenting data in aggregate form only. Survey responses were collected over a 2-month period (September–November 2019). Following this data collection period, final status of NBS for Pompe disease and MPSI were confirmed with NewSTEPs in March 2020 (NewSTEPs, Available online: https://www.newsteps.org/, accessed on 24 March 2020) to determine if any additional states had begun screening for these disorders since the end of our data collection period. The survey contained 57 multiple choice and free-text questions that identified state, role of respondent in NBS, current status of screening for Pompe disease, MPSI, and other LSDs. Additional questions for states who are currently screening for Pompe and/or MPSI including primary analyte, reflex testing, reflex analyte details including use of molecular testing, status of short-term follow-up data collection, and status of long-term follow-up data collection. States currently in the process of planning for NBS of Pompe disease and MPSI were identified and asked questions about plans for primary analyte, reflex testing, reflex analyte details including use of molecular testing, status of short-term follow-up data collection, and status of long-term follow-up data collection. Short-term follow-up describes the data elements needed for state NBS labs to track completion of positive NBS by referral sites (often considered what is needed to ‘close the case’). States currently screening for or planning to start screening for Pompe and MPSI were asked to provide short-term follow-up elements. These answers were divided into three categories: extensive, basic, and genotype only. Extensive short-term follow-up data elements for Pompe disease included detailed clinical information in addition to confirmatory testing (enzyme activity level, urine Hex4, genotype, creatine kinase, transaminases, lactate dehydrogenase, chest X-ray, EKG, echocardiogram, CRIM status, clinical diagnosis type [infantile-vs late-onset], medical management clinic, and barriers to care). Basic short-term follow-up data elements for Pompe disease included confirmatory biochemical testing (enzyme activity), clinical diagnosis type, patient genotype, age at confirmed diagnosis, site of medical management clinic. Extensive short-term follow-up data elements for MPSI included detailed clinical information in addition to confirmatory testing (enzyme activity level, urine glycosaminoglycans, referral for hematopoietic stem cell transplantation, genotype, age at confirmed diagnosis, enzyme replacement therapy start date, medical management clinic, and barriers to care). Basic short-term follow-up elements for MPSI included confirmatory biochemical testing, clinical diagnosis type, patient genotype, age at confirmed diagnosis, site of medical management clinic. Long-term follow-up pertains to any follow-up after initial evaluation and completion of the NBS referral to a clinical site and can last indefinitely into the future. Branching logic was implemented in the survey design for states that were not planning for NBS for Pompe disease or MPSI to not provide them with additional follow-up questions that would only be applicable for states currently doing or in active planning stages for NBS. All questions were optional and respondents were allowed to skip questions if they preferred not to answer them. Thus most responses for detailed follow-up questions do not reflect responses from all 50 states. Data were exported from Qualtrics to Microsoft Excel (Redmond, WA, USA) and RStudio (Boston, MA, USA) for additional analysis and summary. Graphical representations of the data are created in R with the ‘ggplot2′ package [19].

## 3. Results

### 3.1. Current State of Newborn Screening for Lysosomal Storage Diseases in the U.S.

All 50 states responded to our survey. As of March 2020, there are currently 22 states screening for Pompe disease, as shown in Figure 1a. There are 20 states currently screening for MPSI (Figure 1b). In total, 20 states are screening for both LSDs on the RUSP currently. Ten states are screening for other LSDs, including Fabry disease (Illinois, Maryland, Missouri, New Jersey, Oregon, Pennsylvania, Tennessee), Gaucher disease (Illinois, Missouri, New Jersey, Oregon, Pennsylvania, Tennessee), Krabbe disease (Illinois, Kentucky, Missouri, New Jersey, New York, Ohio, Pennsylvania, Tennessee), mucopolysaccharidosis type II (Illinois, Missouri), and Niemann–Pick disease type A/B (Illinois, New Jersey, Pennsylvania).

### 3.2. LSD NBS Planned Expansion by the End of 2020

The majority of states are planning to start screening for Pompe and MPSI by the end of 2020 as shown in Figure 2. In addition to the 22 currently screening, 17 states are planning to start screening for Pompe disease (Figure 1a). Eleven states reported no plans to start screening for Pompe disease. In addition to the 20 states currently screening for MPSI, 20 states are planning to start screening for MPSI. Ten states report no plans to start screening for MPSI (Figure 1b).

### 3.3. LSD NBS by Analyte and Use of Reflex Testing

For both Pompe disease and MPSI, more states reported using or planning to use tandem mass spectrometry (TMS) for measurement of primary analyte (acid alpha-glucosidase or alpha-l-iduronidase activity) compared to digital microfluidics (DMF) platforms (Figure 3a). Fourteen states reported current use of TMS for Pompe disease and nine states are planning to use TMS when Pompe disease screening starts. Seven states currently reported using DMF for Pompe disease and four additional states were considering DMF for Pompe disease screening. Twelve states reported current use of TMS for MPSI and ten states are planning to use TMS when MPSI screening starts. Seven states currently reported using DMF for MPSI and five additional states were considering DMF for MPSI screening. Five and six states reported to be in the planning stages for screening but were undecided on platform for Pompe disease and MPSI, respectively.

Most states currently report use of additional reflex (2nd or 3rd tier) testing on initial NBS sample or are planning to when screening starts (Figure 3b). For Pompe disease, fourteen states report current or planned exclusive molecular testing if the initial NBS is positive. The minority of states have pursued other options including repeating the screen with or without molecular testing if the second NBS is positive or using the CLIR tool (generally for other conditions or specifically for Pompe disease) [11,20] with or without subsequent molecular testing (Figure 3c). For MPSI, eighteen states report current or planned exclusive molecular testing if the initial NBS is positive. Similar to Pompe disease, a smaller fraction of states who are currently or planning to start screening for MPSI use additional biochemical testing (glycosaminoglycans on dried blood spot) [21] with or without molecular testing in addition to repeating the NBS with or without subsequent molecular testing. The majority of reflex molecular testing was performed at commercial labs (eleven states for Pompe disease, ten states for MPSI) compared to state NBS or academic laboratories. If molecular testing is included as reflex NBS testing, cost is included in the NBS fees. Nearly all states currently using molecular testing or planning to use molecular testing report variants of uncertain significance (VUS). Only one state reported consideration of not reporting VUS when NBS starts for MPSI. Discussion of molecular test results and any additional follow-up testing is referred to the medical management clinic.

### 3.4. Practices for Both Short-and Long-Term Follow-Up of Newborn Screening

A portion of the states that are currently screening or planning for screening of Pompe disease and MPSI completed questions on the degree of short-and long-term follow-up that will be collected (Figure 4a,b, respectively). Eight states reported current or planned collection of extensive information on positive NBS for Pompe disease on positive NBS cases. Six states reported current or planned extensive data collection for MPSI. Six states reported basic data element collection for both Pompe disease and MPSI. One state collected genotype only for both Pompe disease and MPSI. The majority of states (nine states for Pompe, eight states for MPSI) reported using an internal state database for long-term follow-up data collection compared to two states that utilized industry-sponsored databases for both Pompe disease and MPSI for long-term follow-up data collection.

## 4. Discussion

Here we present the first snapshot of differences in U.S. states’ practices in NBS for Pompe disease and MPSI in terms of follow-up, primary and reflex analytes. We acknowledge that this was a voluntary survey and that some states completed the survey with varying degrees of detail. We also wanted to ensure the length of our survey was short enough for states to complete as much as possible. Future surveys of U.S. states could address follow-up practices for other LSDs on NBS such as Gaucher, Krabbe, and Fabry diseases. Other questions regarding processes utilized by state labs for decisions related to the choice of primary NBS method (DMF vs. TMS) would be helpful information for all NBS programs to share. Despite this, our survey data reflects an important early step in characterizing NBS practices for LSDs in the United States.

Extensive work has been conducted to decrease NBS false positive rates with implementation of reflex testing [20,21,22,23,24,25]. There are benefits and disadvantages associated with both molecular and biochemical reflex testing. Reflex molecular testing allows for more information regarding phenotype (including CRIM status prediction for Pompe disease) and the presence of pseudodeficiency alleles [26,27]. However molecular testing can also identify VUS that lead to additional interpretation challenges. Biochemical confirmation (enzymatic activity and/or biomarker assays) remains the gold standard for diagnosis, especially in cases where individuals are found to have VUS [28,29,30]. For individuals without insurance or insurance that would not cover genetic testing, molecular reflex testing completed by the NBS state health department could be considered more equitable. Those with insurance may still experience longer turn-around times for separate confirmatory molecular testing and insurance denials [27]. In contrast, biochemical reflex testing provides similar confirmatory and phenotypic information without the inherent challenges associated with VUS interpretation or lack of an individual’s explicit consent for genetic testing [20,27]. While multiple levels of testing reduce the number of false positives, additional reflex testing, including molecular testing, raises concerns that a multi-tier reflex system could create delays for treatment of infantile-onset Pompe disease that leads to worse outcomes. With many states utilizing molecular testing for reflex testing, it is important to consider the added complexity of identifying VUS for families, not just clinicians. Parental anxiety related to false positive or inconclusive NBS has been appreciated for years [31,32], but anxiety related to reporting VUS or pseudodeficiency alleles with exclusive reflex molecular testing may also occur.

Short-term follow-up of positive NBS allows state NBS programs to ensure that the basic goals of screening, detection and treatment initiation, are completed. Long-term follow-up would ideally track confirmed cases detected by NBS over the course of many years and would be able to determine if NBS is cost-effective from a public health perspective. Given the extreme rarity of these disorders, many conditions added to the NBS have limited data on long-term outcomes and efficacy of treatment and would be nearly impossible for individual states to determine. Collaboration between states and nations to track both short-term and long-term outcomes is critical for better understanding these disorders [33,34]. Successful long-term follow-up initiatives require coordinated efforts from multiple groups—including federal agencies, state health departments, academic institutions, primary care providers, and family support organizations [33,35]. As NBS expands to include additional disorders, a redoubled effort for organized long-term follow-up should be considered. Efforts for long-term follow-up at biochemical or genetics clinics have been hampered by the lack of funding for study coordinators’ effort and time for data collection, which may have led towards a greater utilization of industry-sponsored databases that provide funds for clinical research coordinator time for data collection. An ideal situation would evolve towards data collection funded and completed at the public health level. This could be possible if NBS fees included funding for data collection. Even with funding at the state health department level, it would still require data collection at the individual clinic level.

## 5. Conclusions

The first two lysosomal storage disorders (LSDs), Pompe disease and MPSI are now being screened with variable practice in terms of primary and reflex analytes (either biochemical or molecular testing) as well as collection of short-and long-term follow-up elements. Most U.S. states are not currently screening as of March 2020, but this number will nearly double in the next few years. As additional states expand NBS to include these LSDs and more individuals are detected early, it will be imperative to discuss the implementation of further strategies to quantify the outcomes and benefits of NBS for these disorders with consistent collection of short and long term data.

## Figures and Tables

**Figure 1 IJNS-06-00072-f001:**
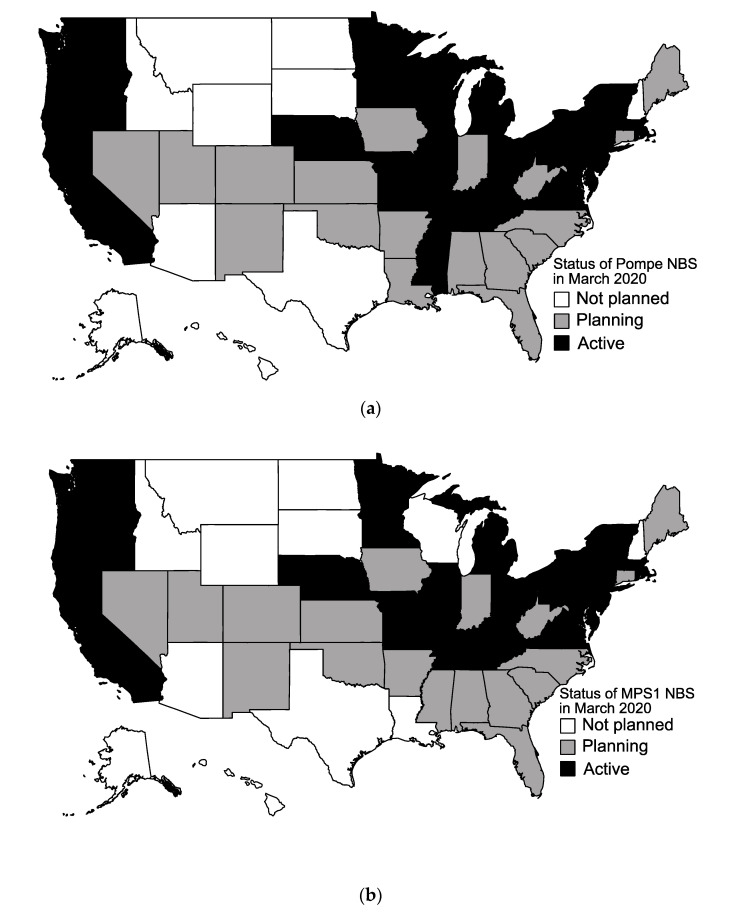
Map of current status of newborn screening (NBS) for Pompe disease and Mucopolysaccharidosis type I (MPSI) as of March 2020. (**a**) Active screening (black), planning stages (gray), and no screening planned (white) for Pompe disease; (**b**) Active screening (black), planning stages (gray), and no screening planned (white) for MPSI.

**Figure 2 IJNS-06-00072-f002:**
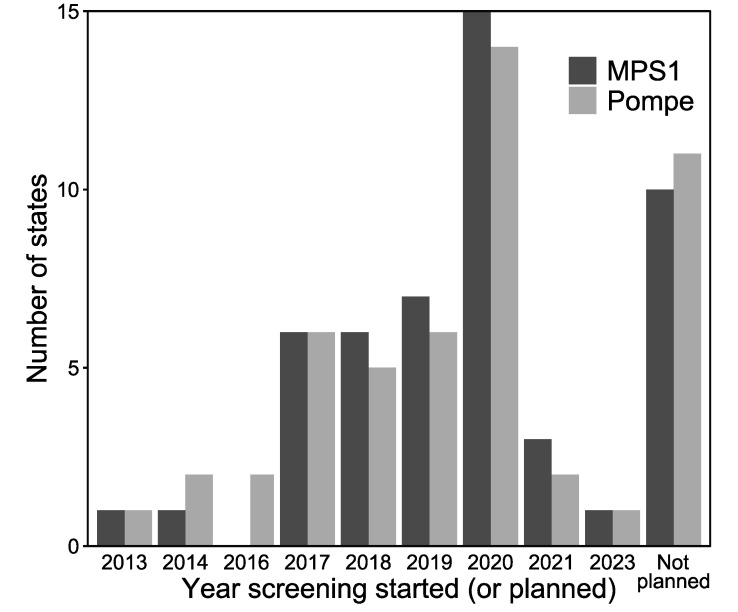
Distribution of when NBS began for MPSI (dark gray) and Pompe disease (light gray) by year.

**Figure 3 IJNS-06-00072-f003:**
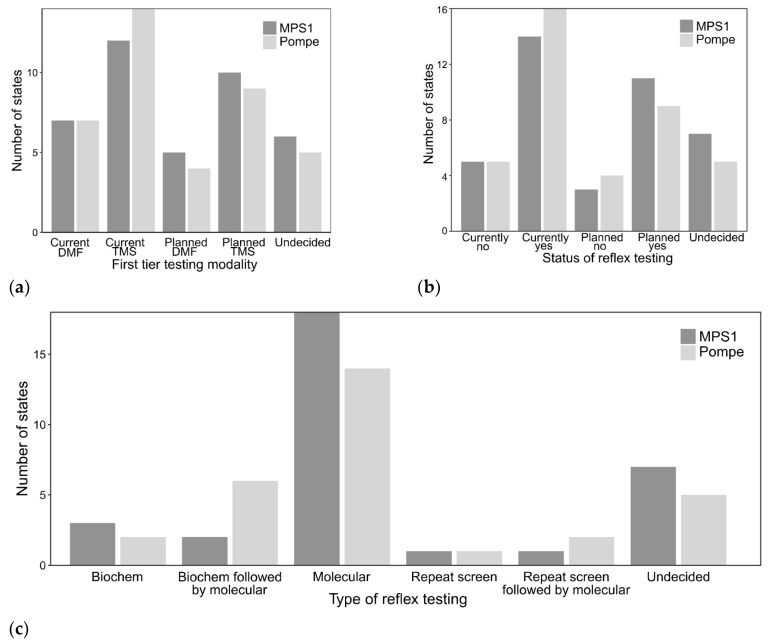
First-tier NBS and reflex testing modality for MPSI (dark gray) and Pompe disease (light gray). (**a**) First-tier testing modality divided between current screening vs. planned screening with either tandem mass spectrometry (TMS) and digital microfluidics (DMF); (**b**) Distribution whether states currently or plan to utilize reflex testing; (**c**) Modality of reflex testing (current and planned screening grouped together) based on additional biochemical testing (Biochem), additional biochemical testing followed by molecular testing (Biochem followed by molecular), molecular testing, repeat screening, repeat screening followed by molecular, and undecided.

**Figure 4 IJNS-06-00072-f004:**
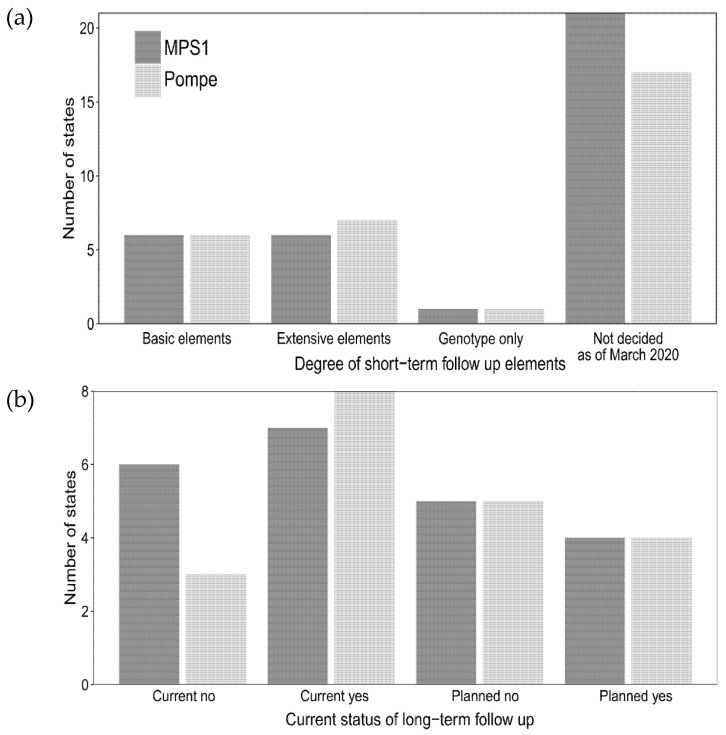
Current and planned efforts for short- and long-term follow-up for MPSI (dark gray) and Pompe disease (light gray). (**a**) Degree of short-term data elements collected (see methods section for explanation of basic and extensive elements); (**b**) Status of long-term follow-up for current and planned screening.

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
