# Peer review of "Current Practices for U.S. Newborn Screening of Pompe Disease and MPSI"

_2409-515X, 2020, doi:10.3390/ijns6030072_

Round 1
Reviewer 1 Report
Thank you for giving me the opportunity to review this interesting article by Ames et.al. The authors provided insightful current status and future plans of newborn screening, reflex, and follow-up testing for Pompe and MPS I diseases. The manuscripts and figures were clearly illustrated.
Line 54:
If some states were contacted by phone and surveys were collected verbally, how to keep it anonymous?
Line 71-78:
It will be very interesting to collect the same elements and survey data for other lysosomal storage diseases such as Fabry, Gaucher, and Krabbe diseases.
Line 92-94:
Have we understood that what are the main factors that NBS labs considered for choosing DMF or TMS as their primary NBS screening method? (ex: The accessibility of the source of the substrates? The cost? The complexity and labor of performing the tests?) If TMS is chosen, the instruments can be also used as second-tier biochemical assays.
Line 102-116:
If most states chose/will choose molecular as reflex testing, where and who will perform the testing (newborn screening labs or referral hospitals or commercial lab)? Where the funding comes from? (NBS labs? Parents? Insurance? Governments’?) Will the genetic testing report directly to the parents? Who will take the responsibility of explaining the VUS or genetic results to the parents?
Line 154-161:
It is great to point out the pros and cons of using molecular or biochemical as reflexing testing. The molecular testing may be more straightforward for most cases but it would cost more and take longer time than biochemical testing and may delay the treatment, especially for the timely manner IOPD cases. Molecular testing has its own limitation especially for the cases with VUS, or VUS combining with one or more pathogenic mutations, or pseduodeficiency variants. The importance of biochemical testing such as leukocyte enzyme activities or biomarker assays should be pointed out for NBS labs and illustrated to readers. Most of the cases with complicated genetic results should be still followed up or interpreted by performing simultaneous biochemical assays.
Line 41 & 168-170:
Several Taiwanese groups have published the papers regarding long-term outcomes and efficacy of treatment for Pompe and MPS diseases. They have launched NBS and comprehensive follow-ups for decades and will be good examples to be referenced here.
Author Response
Dear Reviewer #1,
Thank you for reading and reviewing our manuscript. We hope that with our edits and responses below, our manuscript will be considered for publication. Here are our responses to your concerns:
Line 54: If some states were contacted by phone and surveys were collected verbally, how to keep it anonymous?
We have made the changes requested (page 2, line 60) to reflect that we did know the identity of contacts who provided answers but allowed respondents to maintain anonymity in our reported data by presenting data in aggregate form.
Line 71-78: It will be very interesting to collect the same elements and survey data for other lysosomal storage diseases such as Fabry, Gaucher, and Krabbe diseases.
We very much agree that this information will be very interesting. Due to completion of the data collection period and to ensure the length of the survey was feasible for respondents to complete we chose to currently focus on the LSD’s included on the RUSP. However, this important comment has been included in the discussion section of the manuscript as a limitation of the current survey (page 7, line 183).
Line 92-94: Have we understood that what are the main factors that NBS labs considered for choosing DMF or TMS as their primary NBS screening method? (ex: The accessibility of the source of the substrates? The cost? The complexity and labor of performing the tests?) If TMS is chosen, the instruments can be also used as second-tier biochemical assays.
This is an excellent question, but was unfortunately outside the scope of our survey. These questions were not addressed in our survey; therefore, no changes were made to the manuscript related to this question. However, these very important comments have been included in our discussion as future questions (page 7, line 183).
Line 102-116: If most states chose/will choose molecular as reflex testing, where and who will perform the testing (newborn screening labs or referral hospitals or commercial lab)? Where the funding comes from? (NBS labs? Parents? Insurance? Governments’?) Will the genetic testing report directly to the parents? Who will take the responsibility of explaining the VUS or genetic results to the parents?
These were our main questions regarding states who performed molecular as reflex testing. We have made the changes requested to highlight where molecular testing will be performed on page 5, line 142. Since current states performing molecular testing are including this as part of reflex testing, the associated costs would be included in the NBS fees. We also asked questions regarding how results will be handled, particularly for VUS. From a state NBS laboratory perspective, all states who responded stated that VUS would be included in molecular results and that referral sites (the clinic) would be responsible for follow-up of genetic testing results including disclosure to the family (page 5, line 147). We find this practice important to highlight and could create difficult scenarios and tried to highlight these exact questions in our discussion (page 7, line 193).
Line 154-161: It is great to point out the pros and cons of using molecular or biochemical as reflexing testing. The molecular testing may be more straightforward for most cases but it would cost more and take longer time than biochemical testing and may delay the treatment, especially for the timely manner IOPD cases. Molecular testing has its own limitation especially for the cases with VUS, or VUS combining with one or more pathogenic mutations, or pseduodeficiency variants. The importance of biochemical testing such as leukocyte enzyme activities or biomarker assays should be pointed out for NBS labs and illustrated to readers. Most of the cases with complicated genetic results should be still followed up or interpreted by performing simultaneous biochemical assays.
Wonderful point. We have made the changes requested to highlight the importance of biochemical testing especially in cases when molecular results are unable to confirm the diagnosis (page 7, line 192).
Line 41 & 168-170: Several Taiwanese groups have published the papers regarding long-term outcomes and efficacy of treatment for Pompe and MPS diseases. They have launched NBS and comprehensive follow-ups for decades and will be good examples to be referenced here.
Agreed. We have made the changes requested to provide additional citations for the long-term outcomes for both Pompe disease and MPS1 (page 1, lines 43 and 47, please note that references were added but not flagged as part of track changes).
Reviewer 2 Report
This article describes a snapshot of state newborn screening programs that identify Pompe and MPSI in the U.S.. The authors evinced a thorough effort to gather information from disparate state screening programs and to summarize their findings coherently--not an easy task. This is a descriptive study that is useful to newborn screening programs and to clinicians as a glimpse in time of the screening resources in place for two recently-added RUSP disorders.
The strength of the article is the descriptive power of the data collected about who is screening, what methods individual states use for screening and what kind of follow-up data the states may provide. This is crucial data and the paper has potential for a good overview of it. Where the authors stick to providing adequate descriptions of what they have collected, they did well. Overall, the article needs to be more closely edited for clarity and style and to smooth some awkward sentences. The greatest weakness of the article a lack of well-organized and important definitions of the work that might support the discussion points.
Some examples of editing:
lines 22-23 the abstract is missing a part of the sentence. Lines 168-169, sentence is confusing, "many additions to NBS have". Lines 174-176, sentence is unclear--what is meant by 'coordinator' and how do industry sponsored-databases provide funds?
Clarifications:
1) Please include a brief description of Pompe and MPSI and why early intervention is critical for Pompe. Also, define terms before before using acronyms such as HSCT. Please define pseudodeficiency--both screening professionals and clinicians seem to be unclear about this term.
2) Please state the response rate explicitly at the beginning of the results section. The response rate to the survey seems to be 100% (50 states), then the authors confirmed or updated some screening information by referencing the Newsteps website in March. What information was updated? Was there truly a 100% response?
3) Line 124-138 and figure 3. It is not clear what is meant by basic vs. extensive data elements and short- and long-term data collection. These need to be defined in the methods or placed in a table, not entered parenthetically.
4) Figure 4 Include the numbers of states who responded for the graphs if less than 50. For example, what is the total source numbers for 4c, given the responses in 4b.
Discussion section.
The discussion veers away from the purpose of the article which is to paddle calmly through a difficult stream of information on current state practices. Here we enter into the deep and slippery waters of molecular testing, VUS, timeliness and insurance coverage. I think the opinion piece is out of place in this article, inadequate to the topic, and unnecessary.
Discussion of short- and long-term follow-up is appropriate here, but requires better definition (earlier) and a why. Why collect all that information? I can think of a few reasons, but these need to be fleshed out and stated succinctly. The why will direct what is collected, when it is collected and who might pay for the collection. The authors may have a valuable viewpoint here but I am waiting to hear it.
Author Response
Dear Reviewer #2,
Thank you for reading and reviewing our manuscript. We hope that with our edits and responses below, our manuscript will be considered for publication. Here are our responses to your concerns:
Lines 22-23 the abstract is missing a part of the sentence.
We have made the changes requested to correct this sentence (page 1, line 22).
Lines 168-169, sentence is confusing, "many additions to NBS have".
We have made the changes requested to correct this sentence (page 7, line 212).
Lines 174-176, sentence is unclear--what is meant by 'coordinator' and how do industry sponsored-databases provide funds?
We have made the changes requested to clarify this sentence (page 7, line 220).
Please include a brief description of Pompe and MPSI and why early intervention is critical for Pompe.
This is provided in the Introduction (page 1, line 42) and again highlighted in the discussion (page 7, line 200).
Define terms before before using acronyms such as HSCT.
We have made the changes requested to define this term (page 1, line 44).
Please define pseudodeficiency--both screening professionals and clinicians seem to be unclear about this term.
We have made the changes requested to define this term (page 1, line 37).
Please state the response rate explicitly at the beginning of the results section. The response rate to the survey seems to be 100% (50 states), then the authors confirmed or updated some screening information by referencing the Newsteps website in March. What information was updated? Was there truly a 100% response?
We have made the changes requested to clarify that all 50 states replied to our survey (page 3, line 101). In some cases, states chose not to fully complete the survey, i.e. choice of NBS platform. If additional data was available on NewSTEPs, it was added to our data set for completeness. We have also updated our methods section to clarify this (page 2, line 64).
Line 124-138 and figure 3. It is not clear what is meant by basic vs. extensive data elements and short- and long-term data collection. These need to be defined in the methods or placed in a table, not entered parenthetically
We have made the changes requested to provide additional details in the methods section (page 2, line 73).
Figure 4 Include the numbers of states who responded for the graphs if less than 50. For example, what is the total source numbers for 4c, given the responses in 4b.
We have made the changes requested to clarify these numbers (page 2, line 95). As stated in the Methods section, states were able to choose not to answer any and all questions.
The discussion veers away from the purpose of the article which is to paddle calmly through a difficult stream of information on current state practices. Here we enter into the deep and slippery waters of molecular testing, VUS, timeliness and insurance coverage. I think the opinion piece is out of place in this article, inadequate to the topic, and unnecessary.
We agree that issues like molecular testing, handling of VUS, and insurance coverage are controversial, but are important considerations for some practitioners and state programs. These issues were also specifically brought up as important topics by reviewer #1 and we hope that the manuscript can satisfy all reviewers and be helpful for IJNS readers.
Discussion of short- and long-term follow-up is appropriate here, but requires better definition (earlier) and a why. Why collect all that information? I can think of a few reasons, but these need to be fleshed out and stated succinctly. The why will direct what is collected, when it is collected and who might pay for the collection. The authors may have a valuable viewpoint here but I am waiting to hear it.
We have made the requested changes to define short- and long-term follow-up earlier in the manuscript (page 2, line 73) and to address in the discussion about why it is valuable (page 7, line 208).
Reviewer 3 Report
Overall, I think this manuscript is well written and will be useful for state labs and physicians involved in newborn screening for LSDs.
I would like to know specifically what each state is currently doing. That data is not provided. I would really like to see a table with specific data for each state that is currently screening for Pompe disease and/or MPS1, outlining the year they started, the method of screening they use, use of reflex testing, type of reflex testing and follow-up plans, etc. I have difficulty looking at the figures they way they are shown, and needing to go back and forth to read the text to see the actual numbers.
Author Response
Dear Reviewer #3,
Thank you for reading and reviewing our manuscript. We hope that with our edits and responses below, our manuscript will be considered for publication. Here are our responses to your concerns:
We agree that a table could be a helpful way to present the data. We conducted this survey with the promise that data would be presented in aggregate form and that we would maintain anonymity in our reported data. Many states were initially hesitant to provide answers given that they were still in planning stages of screening and had not completely decided on specifics of NBS such as handling of VUS, but with the guarantee of anonymity, were happy to provide additional information. Data visualizations (both tabular and map-based) are available on NewSTEPs website for disorders screened and, in some cases, primary and reflex analytes (https://www.newsteps.org/data-resources/reports/screening-methodologies-and-targets-report).
Reviewer 4 Report
The manuscript is well written and the objectives and findings are clearly stated. Although deceptively a simple question, the manuscript highlights the high variability in the use of technology and second tier testing across NBS labs in the USA. This is valuable information and a call for uniformity in NBS.
Author Response
Thank you!
Round 2
Reviewer 2 Report
Reads much better. The authors added necessary descriptions and explanations.
Excellent now.